

# Intermittency Analysis of Toy Monte Carlo Events

**Sheetal Sharma and Ramni Gupta**⋆

Department of Physics, University of Jammu, India

⋆ Ramni.Gupta@cern.ch

## Abstract

**Event-by-event intermittency analysis of Toy Monte Carlo events is performed in the scenario of high multiplicity events as is the case at recent colliders RHIC and LHC for AA collisions. A power law behaviour of Normalized Factorial Moments (NFM), $F_q$ as function of number of bins ($M$) known as intermittency, is a signature of self-similar fluctuations. Dependence of NFM on the detector efficiencies and on the presence of fluctuations have been studied. Results presented here provide a baseline to the experimental results and clarity on the application of efficiency corrections to the experimental data.**

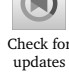

## 1 Introduction

Localized fluctuations in the charged particle production at LHC are proposed to be studied to characterize the multiparticle production and the quark-hadron phase transition [1]. QCD predicts large dynamical fluctuations of various measureables as one of the signatures of critical point, quark-hadron and hadron-quark phase transition. A study of spatial patterns of the charged particles in the phase space using normalized factorial moments (NFM) is one of the techniques to characterize phase transition and the multiparticle production mechanism [2–4]. Normalized factorial moments ($F_q$) of bin multiplicities as function of varying bin size resolution are proposed to be studied for q ≥ 2 [1]. For dynamical fluctuations $F_q > 1$ and is observed to show power-law behaviour with increasing M for self similar fluctuations and this phenomenon is known as intermittency. Here intermittency analysis is performed for the Toy Monte Carlo (ToyModel) events as baseline study.

## 2 Method of Analysis

A sample of 250K high multiplicity Toy Monte Carlo events are generated with two parameters for the tracks corresponding to pseudorapidity ($\eta$) and azimuthal angle ($\phi$) such that $|\eta| \leq 0.8$

and $0 \leq \phi \leq 6.28$. Intermittency analysis (as in [5]) is performed in two dimensional $(\eta, \phi)$ phase space partitioned in $M \times M$ bins, with M $= 4$ to 82. The $q^{th}$ order normalized factorial moment $(F_q)$ is defined as

$$
F_q(M) = \frac{\frac{1}{N} \sum_{e=1}^{N} \frac{1}{M} \sum_{i=1}^{M} f_q(n_{ie})}{\left( \frac{1}{N} \sum_{e=1}^{N} \frac{1}{M} \sum_{i=1}^{M} f_1(n_{ie}) \right)^q} \, ,
\tag{1}
$$

where $f_q(n_{ie}) = \Pi_{j=0}^{q-1}(n_{ie} - j)$, $n_{ie}$ is the bin multiplicity in the $i^{th}$ bin of $e^{th}$ event. $q \geq 2$ is order of the moment and takes positive integer values. $F_q(M)$ shows power law dependence on M as $F_q(M) \propto M^{\phi_q}$ with $\phi_q > 0$ in case there are fluctuations in the bin multiplicities [5]. This scaling behaviour is referred to as *intermittency* and $\phi_q$ as intermittency index. With second order NFM (q $= 2$), the sensitivity of this analysis methodology to gauge bin-to-bin fluctuations and the resilience to detector inefficiencies has been studied in the present work.

## 3 Observations

Normalized factorial moments for q $= 2$ are determined for Toy Monte Carlo events (Toy-Model) using Eq.1. It is observed that for all M, $F_2(M) > 1$ (Fig.1, black filled circles). Also $F_2$ values are independent of number of bins (M). Toy Monte Carlo events do not show any scaling behaviour and hence no intermittency.

For sensitivity check of the analysis methodology a modified sample of events is created from the ToyModel events, using two different approaches. In the first method five percent tracks are added randomly in some phase space bins and an equal number of tracks are removed from rest of the region. In the second method, in a similar fashion five percent tracks are added in some phase space bins but no tracks are removed. For both samples so obtained, intermittency analysis is performed and it is observed that $F_2 > 1$. However $F_2$ is observed to depend on M (Fig.1). At higher M region $F_2$ shows has linear growth with M. This establishes that intermittency analysis methodology is sensitive to the particle density fluctuations.

Calculations for the observables are affected by the detector effects and hence do not give true value. If $\epsilon_i$ defines the detector efficiency in the $i^{th}$ bin then corrected NFM is taken as

$$
F_q(M) = \frac{\frac{1}{N} \sum_{e=1}^{N} \frac{1}{M} \sum_{i=1}^{M} \frac{f_q(n_{ie})}{\epsilon_i^q}}{\left( \frac{1}{N} \sum_{e=1}^{N} \frac{1}{M} \sum_{i=1}^{M} \frac{f_1(n_{ie})}{\epsilon_i} \right)^q} \, .
\tag{2}
$$

From the ToyModel events, which may be called ToyModel(true), two samples of events equivalent to what is measured by the detectors after applying reconstruction routine are obtained. First sample is created by randomly removing 20% of tracks from the acceptance region of each event. The sample so obtained, say ToyModel(U), is 80% of the ToyModel(true) and has uniform efficiency across the acceptance region. For events with non-binomial type efficiencies, 20% particles are removed from some specific phase space regions of each event. This sample of events, say ToyModel(NU), is also 80% of the original events but with different efficiencies across the acceptance region.

Normalized factorial moments are determined for the three samples of events using Eq.1 and corrected NFM are determined for ToyModel(U) and ToyModel(NU) using Eq.2. It is

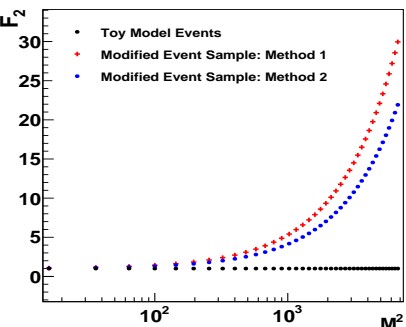

Figure 1: $F_2$ vs $M^2$, depicting sensitivity of analysis technique to gauge bin-to-bin fluctuations in the ToyModel events.

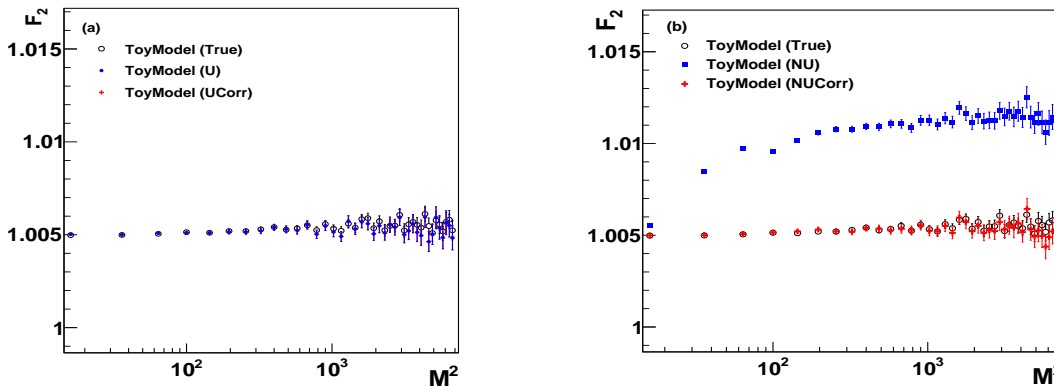

Figure 2: $F_2$ vs $M^2$ plot in case of (a)binomial type efficiencies and (b)non-binomial type efficiencies.

observed that $F_2^{(true)}(M) \approx F_2^{(U)}(M) \approx F_2^{(Ucorr)}(M)$ as is shown in Fig.2(a). However as in Fig.2(b) $F_2^{(true)}(M) \neq F_2^{(NU)}(M)$ (black open circle and blue solid square markers), that is any change in the true track values, introduced differently in different phase space regions, the NFM (Eq.1) do not give true value. Whereas $F_2^{(true)}(M) \approx F_2^{(NUcorr)}(M)$ (Fig.2(b)) implying that with corrected NFM calculated using Eq.2 the true NFM are reproduced. Thus the NFM are robust against binomial detector efficiencies but for non-binomial detector efficiencies, to obtain true NFM, formula (Eq.2) with bin efficiency correction values must be used.

## 4 Conclusions

Intermittency analysis is performed for high multiplicity Toy Monte Carlo events. Analysis technique is observed to be suitable to look for dynamical fluctuations in the multiplicity distributions. NFM as defined in Eq.1 are observed to be robust against the binomial detector efficiencies. However NFM should be corrected for detector effects if efficiencies are non-binomial/non-Gaussian in the acceptance region before any conclusions be drawn.

# Acknowledgements

Author(s) are thankful to Igor Altsybeev, Mesut Arslandok and Tapan Nayak for the discussions and suggesstions that helped in the completion of this study.

**Author contributions**  R.G. designed the model and the computational framework. Both R.G. and S.S. performed the analysis and worked to bring out this manuscript.

**Funding information**  The author(s) received no financial support for the research, authorship and/or publication of this work.

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
