# Peer review of "Intermittency Analysis of Toy Monte Carlo Events"

_SciPost Physics Proceedings, doi:SciPost Phys. Proc. 10, 024 (2022)_

## Round 1 · Referee Report · Anonymous (Referee 1) · 2022-1-27

Report

In this contribution to the ISMD2021 proceedings the authors report on a toy study to assess the resilience of intermittency to detector inefficiencies. The manuscript is suitable for proceedings and meets the requirements, but hard to understand for non-experts. I therefore have a few suggestions that should heko to make it more understandable: 1) In the introduction say fluctuations/NFM of which quantity are considered? I understand it is particle multiplicity, right? 2) Define the symbols (e.g. q, F_q, f_qi^rec). 3) Some technical details like using the system clock as seed for the random number generator and which function in Root was used can be left out to create a bit of space. 4) Second to last sentence before eq. (1): What is meant by 'in the interval of 2'? 5) In section 3: a dependence of F_q on M is expected (cf. eq. (2)), or? In the manuscript it sounds as if this was surprising. 6) Do the efficiencies epsilon depend on eta and phi for the NU sample?

  • validity: -
  • significance: -
  • originality: -
  • clarity: -
  • formatting: -
  • grammar: -

Author:  Sheetal Sharma  on 2022-02-13  [id 2194]

(in reply to Report 1 on 2022-01-27)

Thanks for your comments on the first draft of the proceedings this will certainly help to improve it and convey the scientific message clearly. Apologies for the delay in reply due to some health issues with me. We will update the draft based on your suggestions and upload the second version on arXiv. Please find below our responses to your comments

1) Reply: Yes, you are right it is Normalized factorial moments (NFM) of bin multiplicity. In seventh line after [2,4] we will update it as;

Normalized factorial moments (NFM) of bin multiplicities as function of varying bin size.......

2) Reply: q is the order of the moment, F_q is NFM, we will mention this in the first three lines of Page 2. Yes we missed to mention f_qi^rec.. Will include what f_qi^rec after Eq.(3) as;

Where superscript rec in f_qi^rec corresponds to reconstructed f_qi of reconstructed data in experiment.

3) Reply: Thanks for this suggestion to create space.

4) Reply: It means that we are varying M i.e. number of bins in the phase space from 4 up to 82 in the interval of 2 as 4, 6, 8, 10,......... 82 However its not that important and obvious from figures, we can remove this.

5) Reply: This is mentioned in section 1 that for dynamical fluctuations Fq ≥1 and Fq shows power law dependence on M with increasing M. Like in this case, we are adding fluctuations (by increasing particle densities in certain bins) artificially in the phase space. With Eq (2) we will includes this as:

Fq(M) has power law dependence M as Fq α M^ϕq (ϕq known as intermittency index and is >0) in case there are fluctuations in the bin multiplicity.

6) Reply: Because of shortage of space, everything has to be limited in 3 pages, the definition of efficiency is not much clear from text which is Efficiency = {Number density of bins in Reconstructed sample} over {Number density in the true sample} ϵ or efficiency depends on how robust is the reconstruction technique and/or detector’s capability to detect particles. For a detector which detects/measures uniformly in the whole acceptance regions, where it is effective, the efficiency value will be same and this technique of looking for fluctuations is robust against detector efficiencies in that case.

Here with TMC, to have NU sample for reconstructed data assuming that the detector has not equal probable measurements at all acceptance regions, reconstructed sample is created by taking out tracks unequally i.e. removing track from some specific η, ϕ value so that we get a case of non-uniform or non-binomial ϵ. Thus efficiencies are not function of ( η, ϕ) as such

Attachment:

ReplyToComments.pdf

---

## Round 3 · List of Changes

As suggested by the referee we have made following changes in the manuscript as given below
1.) Introduction:
Seventh line updated as; Normalized factorial moments (NFM) of bin multiplicities as function of .............
2.) Method of Analysis:
First line rearranged as; A sample of 250K high multiplicity Toy Monte Carlo events are generated with two parameters...........
Third line; TRandom3() function of ROOT ........removed to create some space.
Symbols are defined after Eq.(1)and Eq.(2) is now included in the text
Last two - three lines are also modified to make the things more clear
3.) Observations:
Figure. 1 is updated to make the text clear in the first paragraph.
Eq.(3) is now Eq.(2) and this is also modified in terms of Eq.(1). Symbols are also correctly defined.
TMC is now written as ToyModel
4.) Conclusions: No change
1.) Introduction:
Seventh line updated as; Normalized factorial moments (NFM) of bin multiplicities as function of .............
2.) Method of Analysis:
First line rearranged as; A sample of 250K high multiplicity Toy Monte Carlo events are generated with two parameters...........
Third line; TRandom3() function of ROOT ........removed to create some space.
Symbols are defined after Eq.(1)and Eq.(2) is now included in the text
Last two - three lines are also modified to make the things more clear
3.) Observations:
Figure. 1 is updated to make the text clear in the first paragraph.
Eq.(3) is now Eq.(2) and this is also modified in terms of Eq.(1). Symbols are also correctly defined.
TMC is now written as ToyModel
4.) Conclusions: No change

---

## Editorial Decision

published